# Kinetic mechanism of coupled binding in sodium-aspartate symporter GltPh

SeCheol Oh[1†], Olga Boudker[1,2]*

[1]Department of Physiology & Biophysics, Weill Cornell Medicine, Cornell University, New York, United States; [2]Howard Hughes Medical Institute, Chevy Chase, United States

**Abstract** Many secondary active membrane transporters pump substrates against concentration gradients by coupling their uptake to symport of sodium ions. Symport requires the substrate and ions to be always transported together. Cooperative binding of the solutes is a key mechanism contributing to coupled transport in the sodium and aspartate symporter from *Pyrococcus horikoshii* Glt_Ph. Here, we describe the kinetic mechanism of coupled binding for Glt_Ph in the inward facing state. The first of the three coupled sodium ions, binds weakly and slowly, enabling the protein to accept the rest of the ions and the substrate. The last ion binds tightly, but is in rapid equilibrium with solution. Its release is required for the complex disassembly. Thus, the first ion serves to 'open the door' for the substrate, the last ion 'locks the door' once the substrate is in, and one ion contributes to both events.

DOI: https://doi.org/10.7554/eLife.37291.001

*For correspondence: olb2003@med.cornell.edu

**Present address:** †Structural Biology Program, Memorial Sloan Kettering Cancer Center, New York, United States

## Introduction

Glt_Ph is an archaeal homologue of glutamate transporters (excitatory amino acid transporters, EAATs), which move the neurotransmitter from the synaptic cleft into the cytoplasm of glial cells and neurons, thereby terminating neurotransmission events (*Danbolt, 2001*). Glutamate transport by EAATs is driven by symport of three sodium (Na⁺) ions and one proton and by antiport of a potassium ion (*Zerangue and Kavanaugh, 1996*; *Levy et al., 1998*; *Owe et al., 2006*). Under pathophysiologic conditions such as ischemia or stroke, ionic gradients across membranes dissipate, and EAATs operate in reverse, releasing glutamate from the cells. Sustained elevated glutamate concentrations lead to excitotoxicity and ultimately to cell death and brain damage (*Yi and Hazell, 2006*; *Sheldon and Robinson, 2007*).

Glt_Ph, which shares ~ 35% amino acid sequence identity with EAATs, has served as a structural and mechanistic model for this family of transporters (*Yernool et al., 2004*; *Boudker et al., 2007*; *Reyes et al., 2009*; *Reyes et al., 2013*; *Akyuz et al., 2013*; *Akyuz et al., 2015*; *Verdon et al., 2014*; *Scopelliti et al., 2018*). Like EAATs, it symports the substrate aspartate (L-asp) together with three Na⁺ ions (*Figure 1—figure supplement 1*), but is independent of protons or potassium ions (*Ryan et al., 2009*; *Groeneveld and Slotboom, 2010*; *Boudker et al., 2007*). Structural studies on Glt_Ph, a closely related Glt_Tk, and human EAAT1 showed that these proteins are homotrimers (*Yernool et al., 2004*; *Canul-Tec et al., 2017*; *Guskov et al., 2016*). Each protomer consists of a peripheral transport domain, which is a mobile unit harboring the substrate- and ion-binding sites, and a rigid central scaffold or trimerization domain (*Figure 1—figure supplement 1*) (*Reyes et al., 2009*). The conformational transitions between the outward- and inward-facing states involve translocation of the transport domain across the membrane by ~ 15 Å in a stochastic, thermally driven process that occurs independently in individual subunits (*Reyes et al., 2009*; *Akyuz et al., 2013*; *Hänelt et al., 2013*; *Erkens et al., 2013*; *Georgieva et al., 2013*; *Ruan et al., 2017*; *Verdon and Boudker, 2012*).

When substrate binds to the transporter in the outward-facing state, a helical hairpin (HP) two acts as a gate (*Verdon et al., 2014*; *Akyuz et al., 2013*; *Focke et al., 2011*; *Jensen et al., 2013*). In the apo transporter and in the transporter bound to both L-asp and Na$^+$ ions, HP2 closes down over the substrate-binding site, but it remains open when the transporter is bound to Na$^+$ ions only (*Figure 1—figure supplement 2*). Inability of HP2 to close in the absence of substrate may prevent uncoupled ion transport (*Verdon et al., 2014*). HP2 assumes an even more open conformation in the structures of Glt$_{Ph}$ and EAAT1 bound to a blocker DL-*threo*-β-benzyloxyaspartic acid (DL-TBOA) and to other non-transportable inhibitors (*Boudker et al., 2007*; *Canul-Tec et al., 2017*; *Scopelliti et al., 2018*). The conformations of the apo and substrate-bound transport domains are identical in the outward- and inward-facing states (*Reyes et al., 2009*; *Verdon et al., 2014*), though what constitutes a gate in the latter state is unclear. HP1, related to HP2 by the structural pseudo-symmetry, was considered as a possible gate (*Reyes et al., 2009*), but it lacks the abundance of gly-cine residues that confers flexibility to HP2 (*Verdon et al., 2014*). HP2 itself is constrained at the interface between the transport domain and the scaffold. Nevertheless, inward-facing Glt$_{Ph}$ binds DL-TBOA and the structurally related inhibitor L-*threo*-β-benzylaspartic acid (L-TBA), suggesting that a conformation with an open HP2 exists (*Reyes et al., 2013*). Consistently, a crystal structure of an inward-facing mutant R276S/M395R, dubbed 'fast mutant' because it transports L-asp about four times faster than the wild type Glt$_{Ph}$ (*Ryan et al., 2010*), pictured the transport domain leaning away from the scaffold leaving sufficient space for HP2 to open (*Akyuz et al., 2015*).

The linchpin of the coupled transport in Glt$_{Ph}$ is the cooperative binding of L-asp and three Na$^+$ ions, such that alone the solutes bind with low affinity, but together they bind tightly (*Reyes et al., 2013*). Both outward- and inward-facing transporter shows this property as has been demonstrated using conformationally constrained variants. These were generated by cross-linking double cysteine mutants strategically placed into the scaffold and transport domains (*Reyes et al., 2013*). Conse-quently, when the substrate-binding site faces the extracellular solution containing high Na$^+$ concen-tration, the substrate affinity is high and when the site is exposed to the low cytoplasmic Na$^+$ concentrations, the affinity drops and the substrate is released. While the thermodynamics of bind-ing is well characterized, what is the kinetic role of Na$^+$ ions is less well understood. Equilibrium stud-ies showed that at least two Na$^+$ ions are able to bind weakly to both outward- and inward-facing transporter in the absence of L-asp. Therefore, it was postulated that their binding preceded L-asp binding (*Reyes et al., 2013*). Consistently, kinetic studies on conformationally unconstrained Glt$_{Ph}$ variants showed that binding of Na$^+$ ion or ions preceded binding of the substrate (*Hänelt et al., 2015b*; *Ewers et al., 2013*). Notably, these studies were difficult to interpret mechanistically because reorientation of the transport domain might have contributed to the observed events. Nev-ertheless, they likely approximated binding in the outward-facing state because the unconstrained Glt$_{Ph}$ samples this state preferentially (*Georgieva et al., 2013*; *Akyuz et al., 2013*; *Ruan et al., 2017*).

Here we describe a kinetic mechanism underlying the thermodynamically coupled binding and release of three Na$^+$ ions and L-asp in the inward-facing state of the transporter. Using a tryptophan mutant of the conformationally constrained inward facing transporter, we find that both binding and dissociation of L-asp are slow and controlled by Na$^+$ ions. Specifically, we show that binding of the first Na$^+$ ion is rate-limiting to the complex formation. We further show that once the complex is assembled, at least one Na$^+$ ion is in rapid equilibrium with solution, and its dissociation is pre-requi-site for the complex disassembly. Based on these findings, we construct a minimal kinetic model and show that there are no unique binding and disassembly mechanisms, but rather probabilistically weighted reaction paths. Which path is taken is controlled by Na$^+$ concentration in solution. How-ever regardless of the path, combination of the very slow weak binding of the first Na$^+$ ion and very rapid and tight binding of the last Na$^+$ ion ensures that neither equilibrium nor kinetic intermediates are populated during Na$^+$ and L-asp unloading and loading, provided both solutes are encountered by the transporter at the same time.

## Results

### Na⁺ ions control binding and unbinding rates of L-asp

To detect ligand binding to the inward-facing state of Glt$_{Ph}$, we introduced a tryptophan mutation, P11W in the trans-membrane (TM) segment 1 of the scaffold domain within the context of Hg$^{2+}$-crosslinked K55C/C/321A/A364C mutant (Glt$_{Ph}$$^{IFS}$), referred to as P11W$^{IFS}$ (*Figure 1a*). Notably, Glt$_{Ph}$ contains no native tryptophans. The fluorescence of P11W increases upon coupled binding of Na⁺ ions and L-asp or blocker DL-TBOA (*Figure 1b* and *Figure 1—figure supplement 3*). The affinity of P11W$^{IFS}$ for L-asp was similar to that of the wild type Glt$_{Ph}$$^{IFS}$ over a range of Na⁺ concentrations (*Figure 1c*). The slope of the log-log plot of L-asp dissociation constants as a function of Na⁺ concentration was 2.5, similar to that of the wild type transporter, suggesting a nearly complete coupling between L-asp and three Na⁺ ions. Taken together, these results show that the binding process is not significantly affected by P11W mutation.

The tight thermodynamic coupling has to manifest in strong dependence of L-asp binding on- and/or off-rates on Na⁺ concentration. If all three Na⁺ ions were coupled to L-asp through accelerating binding, the L-asp binding rate would show cubic dependence on Na⁺ concentration. Upon mixing the transporter with Na⁺ ions and L-asp, the equilibration process would be described by a single exponential yielding a rate constant, $k_{obs}$:

$$k_{obs} = k_{off} + k_{on}[Na^+]^3[asp] \tag{1}$$

where $k_{on}$ and $k_{off}$ are binding and dissociation rate constants, respectively. Addition of L-asp to the apo P11W$^{IFS}$ in the presence of variable concentrations of Na⁺ ions led to a time-dependent increases of fluorescence signal (*Figure 1—figure supplement 3c*) that were well described by single exponentials, yielding $k_{obs}$ values. As expected, the equilibration rates increased with increased Na⁺ concentration (*Figure 2a*), but not as steeply as *Equation 1* requires, showing quadratic dependence at low L-asp concentrations and linear dependence at high L-asp concentrations (*Figure 2—figure supplement 1*). These observations suggest that binding of one or two Na⁺ ions facilitate the amino acid binding.

These data further suggest that the role of Na⁺ ions cannot be limited to accelerating L-asp binding and that the ions should also slow L-asp dissociation. To test this hypothesis, we prepared a P11W$^{IFS}$ stock pre-bound to L-asp and Na⁺ ions. We then diluted the stock to the final concentration of 0.25 µM transporter, 0.25 µM L-asp and Na⁺ ions varying between 50 µM and 1 mM. Under these conditions, L-asp is expected to dissociate from the protein nearly completely as the K$_D$ for L-asp is ∼ 100 µM at 1 mM Na⁺ (*Figure 1c*). Upon dilution, we observed gradual decreases of fluorescence reflecting L-asp and Na⁺ dissociation, which were well described by single exponential decay functions (*Figure 1—figure supplement 3d*). The obtained $k_{obs,d}$ decreased steeply with increased Na⁺ concentrations (*Figure 2b*). This behavior is in contrast to *Equation 1*, which presumes that $k_{off}$ is Na⁺-independent. To explain it, we hypothesized that one or more Na⁺ ions were in rapid equilibrium between the complex and solution and that L-asp dissociated only when the ions were off (*Figure 2c*). If so, the rapidly equilibrating Na⁺ ions would respond to the changes of the ion concentration in the media, and the rate of L-asp dissociation would depend on the fraction of the transporter free of the ions:

$$k_{obs,d} = k_{off,asp} \frac{K_{D,Na}^n}{[Na^+]^n + K_{D,Na}^n} \tag{2}$$

where $k_{off,asp}$ is the intrinsic rate constant of L-asp dissociation once the rapidly equilibrating Na⁺ ions are released, $K_{D,Na}$ is the dissociation constant of the equilibrating Na⁺ ions and $n$ is the Hill coefficient. Hill coefficients between 1 and 3 described data similarly, producing fitted $K_{D,Na}$ values between 30 and 150 µM, respectively. The $k_{off,asp}$ is between 0.5 and 0.2 s$^{-1}$ suggesting that even when the locking Na⁺ ion(s) are released, it still takes approximately 2 to 5 s for the rest of the complex to disassemble.

These experiments demonstrate that as the concentration of Na⁺ ions in the media increases, L-asp release slows down. Consistently, we observed that in proteoliposomes containing Glt$_{Ph}$, the rate of concentrative Na⁺-driven uptake of radiolabeled L-asp into vesicles containing Na⁺-free buffer was significantly higher than the rate of L-asp exchange when both vesicle internal and

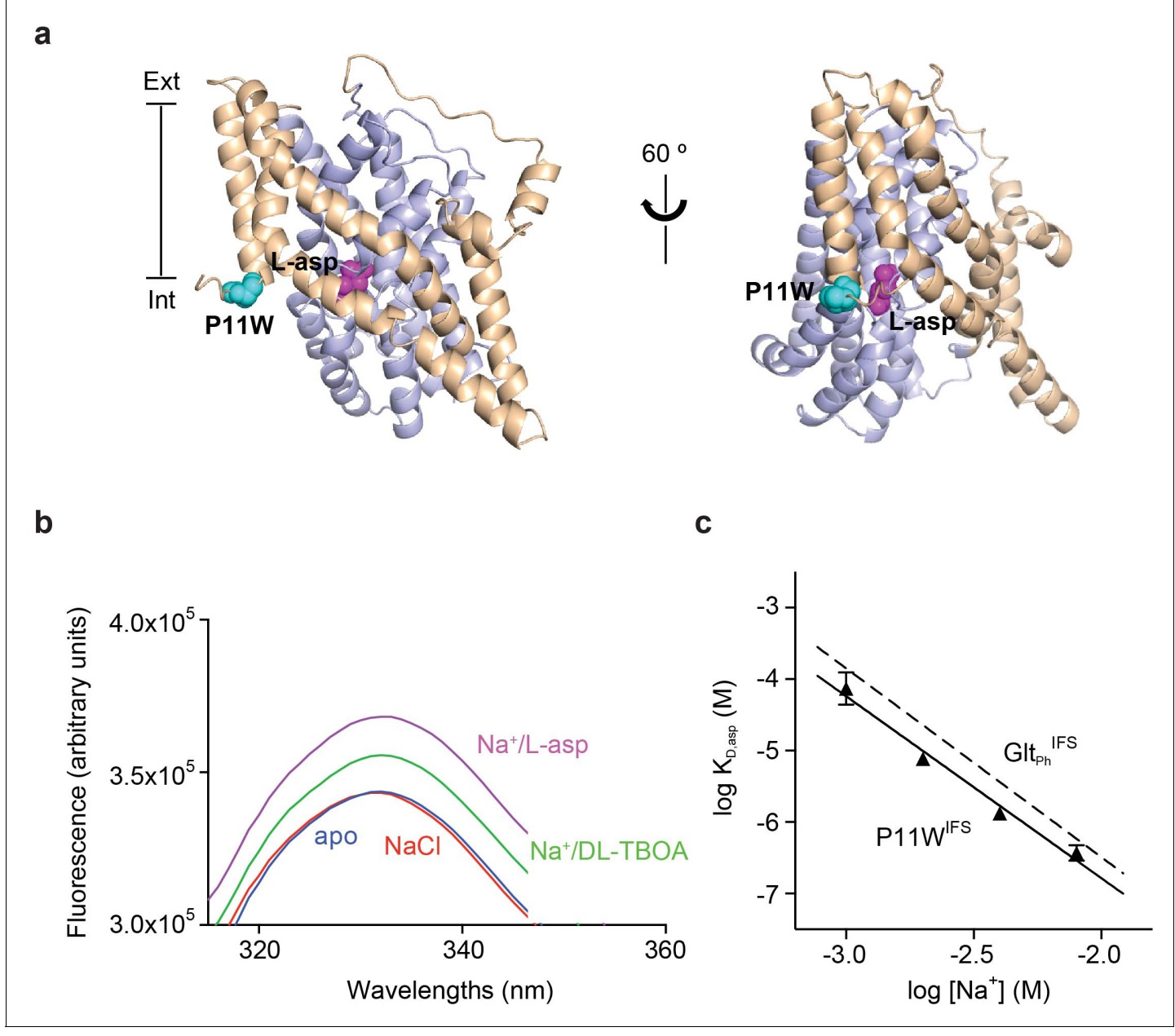

**Figure 1.** Design and characterization of P11W$^{IFS}$. (a) Pro11 (cyan) in the trimerization domain (wheat) of Glt$_{Ph}$$^{IFS}$ (PDB ID: 3KBC) is close to substrate binding site (magenta) in the transport domain (blue). (b) Intrinsic fluorescence emission spectrum of apo P11W$^{IFS}$ (blue) and P11W$^{IFS}$ in the presence of 10 mM NaCl (red) and DL-TBOA (green) or L-asp (magenta). (c) Dependence of L-asp dissociation constant, $K_D$ of P11W$^{IFS}$ on Na$^+$ ion concentration. The data (solid triangles) were fitted to a straight line with slope of 2.5. Binding assay based on RH421 fluorescence was used to follow L-asp binding to P11W$^{IFS}$. The dashed line represents Na$^+$ dependence of L-asp $K_D$ of Glt$_{Ph}$$^{IFS}$ with slope of 2.6 (*Reyes et al., 2013*).

DOI: https://doi.org/10.7554/eLife.37291.002

The following figure supplements are available for figure 1:

**Figure supplement 1.** Na$^+$-coupled L-asp transport by Glt$_{Ph}$.

DOI: https://doi.org/10.7554/eLife.37291.003

**Figure supplement 2.** HP2 acts as a gate for substrate binding.

DOI: https://doi.org/10.7554/eLife.37291.004

**Figure supplement 3.** Characterization of P11W$^{IFS}$ Glt$_{Ph}$.

DOI: https://doi.org/10.7554/eLife.37291.005

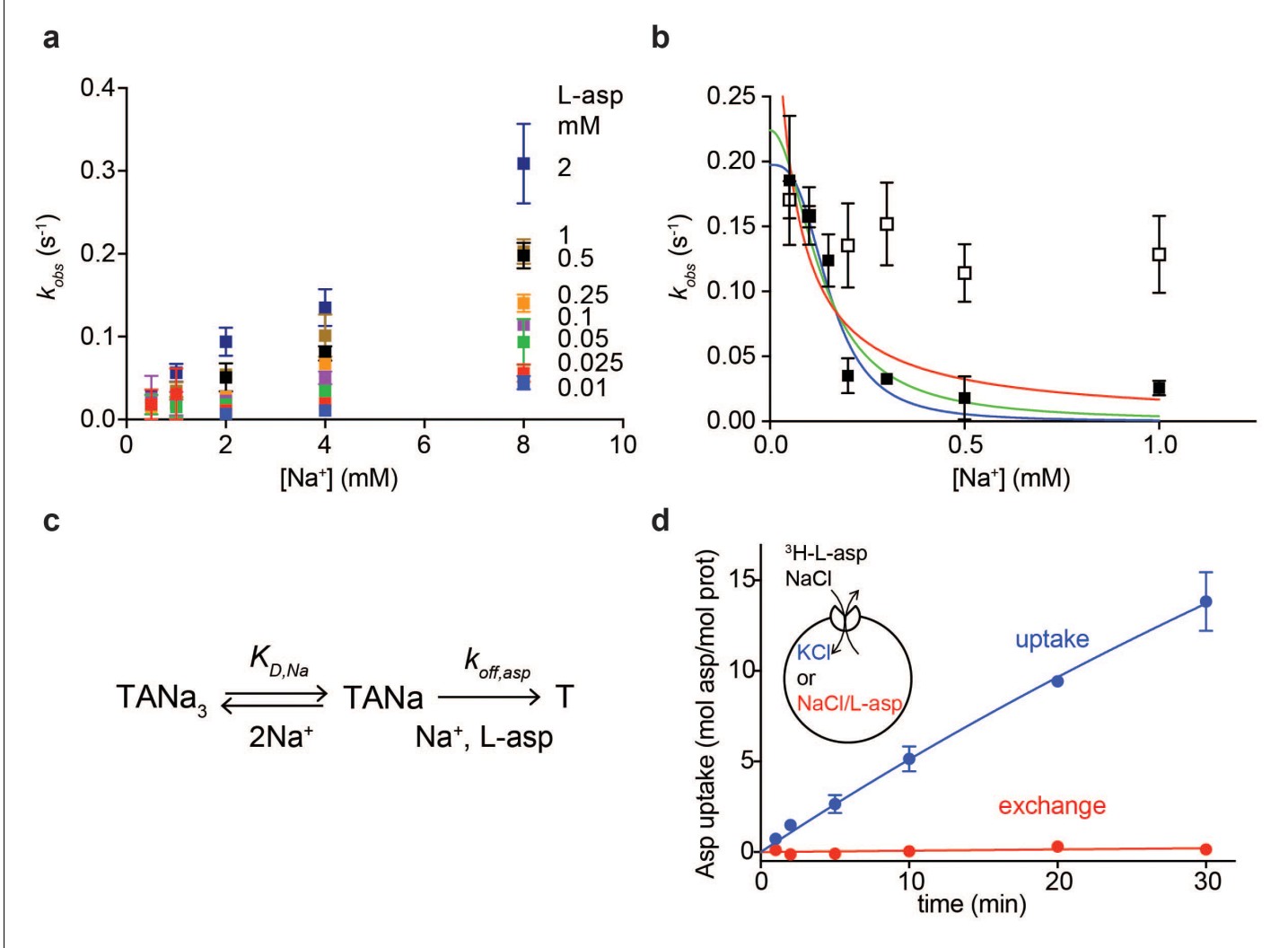

**Figure 2.** Na$^+$ ions control on- and off-rates of L-asp binding. (**a**) The values of $k_{obs}$ for L-asp binding as a function of Na$^+$ concentration. The binding rates were measured as a function of Na$^+$ concentrations upon additions of L-asp to P11W$^{IFS}$ to the following final concentrations in mM: 0.01 (blue), 0.025 (red), 0.05 (green), 0.1 (purple), 0.25 (orange), 0.5 (black), 1 (brown) and 2 (dark blue). (**b**) The dissociation rates of L-asp as a function of Na$^+$ concentration in the media measured for P11W$^{IFS}$ (solid squares) and P11W$^{IFS}$-M311A (open squares). The lines through data for P11W$^{IFS}$ are the fits to **Equation 2** with Hill coefficients $n$ fixed to 1 (red), 2 (green) and 3 (blue). The fitted values of $k_{off,asp}$ and $K_{D,Na}$ were, respectively: $0.5 \pm 0.2$ s$^{-1}$, $40 \pm 20$ μM; $0.2 \pm 0.02$ s$^{-1}$, $130 \pm 20$ μM; $0.2 \pm 0.01$ s$^{-1}$, $150 \pm 13$ μM. (**c**) Schematic representation of the mechanism in **Equation 2** exemplified for two rapidly equilibrating cooperatively binding Na$^+$ ions. (**d**) Comparison of the rate of L-asp uptake (open circles) and exchange (solid circles) in proteoliposomes containing wild type Glt$_{Ph}$. Background uptake was measured in the absence of the external NaCl and subtracted from the data. All experiments were performed in triplicate and means and standard errors are shown.

DOI: https://doi.org/10.7554/eLife.37291.006

The following figure supplement is available for figure 2:

**Figure supplement 1.** Na$^+$ dependence of L-asp binding kinetics in low (**a**) and high (**b**) L-asp.

DOI: https://doi.org/10.7554/eLife.37291.007

external solutions contained L-asp and Na$^+$ ions (**Figure 2d**). In the latter experiment, the proteoliposomes are loaded with high Na$^+$ and cold L-asp that exchanges with radiolabeled L-asp in the external solution. Under these conditions, high Na$^+$ concentrations on both sides of the membrane slow the release of the substrate from the transporter, resulting in a slow exchange process. The Na$^+$-dependent release mechanism is also consistent with kinetic measurements on the mammalian EAAT4, for which the transport cycle was highly sensitive to the cytoplasmic Na$^+$ ions concentrations (**Wadiche et al., 2006**). These results further explain why the measured Km values for Glt$_{Ph}$, ~100 nM

at 100 mM NaCl (*Ryan et al., 2009*), are significantly higher than L-asp $K_D$ of ~ 1 nM measured in detergent solutions: L-asp bound to the transporter is more likely to be translocated into the vesicle and released into Na$^+$-free lumen than to dissociate back into the media containing high Na$^+$ concentration.

Collectively, these experiments show that one or two Na$^+$ ions control the on-rate of L-asp binding and one or more ions control the off-rate. The importance of the Na$^+$-dependent off-rate in cooperative Na$^+$ and L-asp binding is consistent with the observation that introducing M311A mutation, which leads to largely Na$^+$-independent L-asp binding (*Verdon et al., 2014*), into P11W$^{IFS}$ eliminates Na$^+$ dependence of L-asp dissociation rates (*Figure 2b*).

## Na$^+$ ions drive conformational transition into a binding competent state

Plotting equilibration $k_{obs}$ constants as a function of L-asp concentration shows that the dependence is not linear, as *Equation 1* would predict, but that instead $k_{obs}$-s plateau at high L-asp concentrations (*Figure 3a*). Thus, slow conformational transitions independent of L-asp become rate-limiting. Typically, two mechanisms are invoked to explain such behavior: induced fit and conformational selection. According to induced fit mechanism, initial weak binding is followed by a slow isomerization into a high affinity state. In conformational selection, the slow isomerization into a high affinity state precedes binding (*Vogt et al., 2012*). Either of these mechanisms can lead to $k_{obs}$ values that level off with increased ligand concentration approaching a maximum. Notably, the observed plateaus are Na$^+$-dependent; therefore, the rate-limiting step involves Na$^+$ binding.

To further probe the mechanism of complex formation, we examined the kinetics of DL-TBOA binding to P11W$^{IFS}$ because it is thought to recapitulate early events of substrate binding that precede closure of HP2 and Na$^+$ binding to Na2 site (*Figure 1—figure supplement 2*) (*Reyes et al., 2013*). Consistently in equilibrium experiments, binding of L-TBOA to unconstrained Glt$_{Ph}$ and of L-TBA to Glt$_{Ph}$$^{IFS}$ showed reduced Na$^+$ coupling (*Boudker et al., 2007*; *Reyes et al., 2013*). Addition of DL-TBOA to P11W$^{IFS}$ at variable Na$^+$ concentrations led to time-dependent increases of tryptophan fluorescence that were fitted to single exponentials (*Figure 1—figure supplement 3e*). Strikingly, equilibration rate constants decreased with increased concentrations of DL-TBOA, asymptotically approaching minima (*Figure 3b*). Such behavior is a hallmark of conformational selection mechanism (*Vogt et al., 2012*). It occurs because the binding-competent conformation is a minor state sampled by the protein. The higher is the concentration of the added ligand, the larger proportion of the protein slowly isomerizes into the binding-competent state that is then rapidly sequestered by the ligand. Therefore, it takes longer for the system to equilibrate at higher ligand concentrations until saturation is reached. Because the plateau values increased at increased Na$^+$ concentrations, the slow isomerization must be coupled to Na$^+$ binding. Thus, we envision that one or more Na$^+$ ions bind slowly and weakly to the transporter, so that only a small fraction of the protein is bound to the ion alone. Under 'rapid equilibrium approximation', where binding and unbinding of DL-TBOA are assumed to be faster than the preceding conformational changes, associated with Na$^+$ ion binding (*Figure 3c*), we can fit the data to an equation:

$$k_{obs} = k_r + k_{-r} \frac{K_{D,TBOA}}{K_{D,TBOA} + [TBOA]} \tag{3}$$

where $k_r$ and $k_{-r}$ are the forward and reverse rate constants of the conformational change and $K_{D,TBOA}$ is the affinity of the inhibitor for the high affinity state of the transporter. Notably, $k_r$ are Na$^+$-dependent since they reflect Na$^+$ ion binding. The asymptotic values of $k_r$ are resolved by the data, but $k_{-r}$ and $K_{D,TBOA}$ are correlated and could not be determined. Plotted as a function of Na$^+$ concentration, $k_r$ values show linear dependence (*Figure 3d*) consistent with binding of one Na$^+$ ion coupled to the slow conformational transition.

Based on these data, we hypothesized that L-asp and DL-TBOA share the same rate-limiting Na$^+$ binding event that is prerequisite for the amino acid binding. In contrast to DL-TBOA, the rates of L-asp binding increased with ligand concentration. This is expected to be the case when the rate of L-asp dissociation, $k_{off}$ is slower than $k_{-r}$ (*Figure 3c*). In this case 'rapid equilibrium approximation' is not valid, and the data must be described by a full equation (*Vogt et al., 2012*):

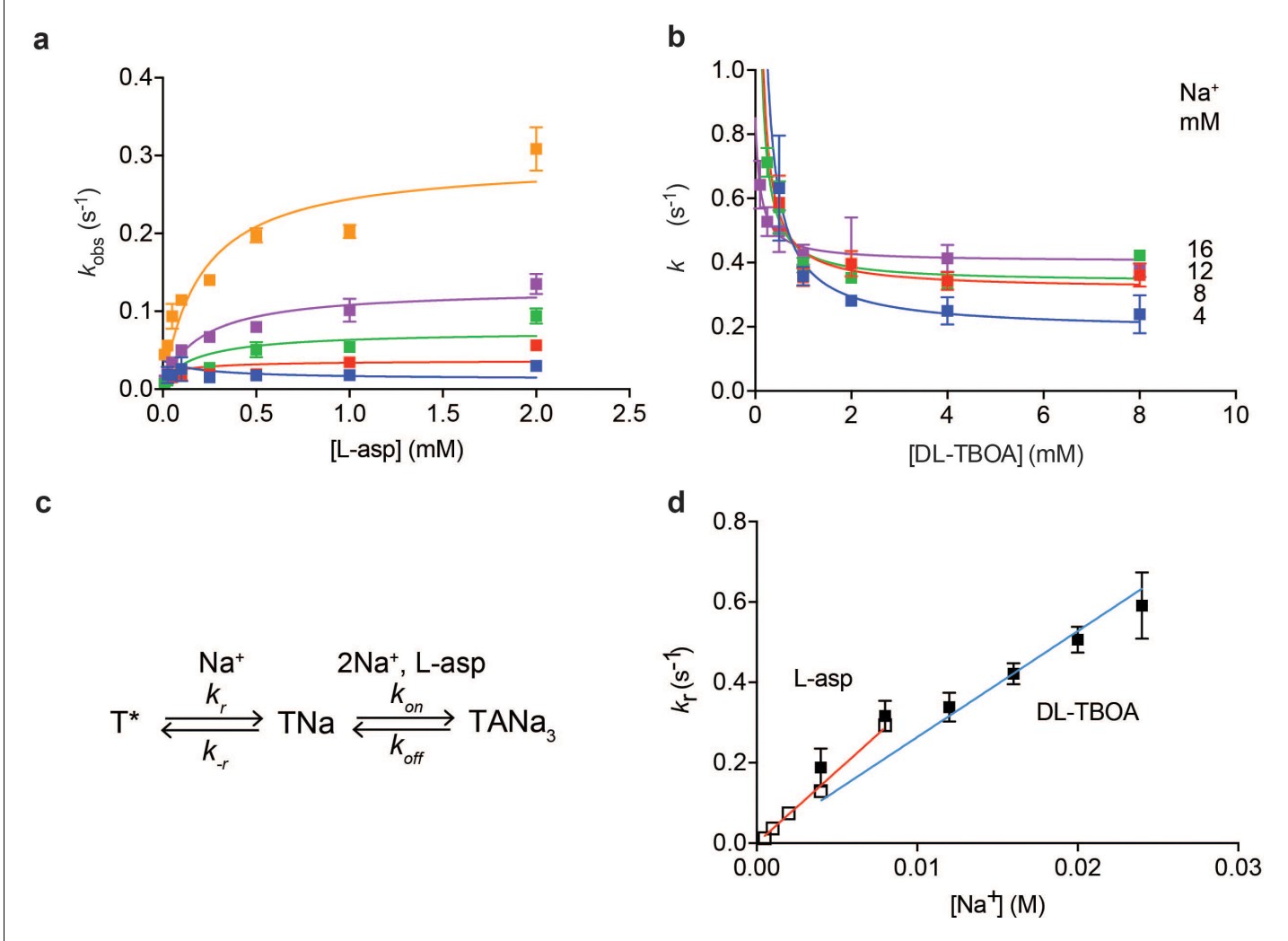

**Figure 3.** Na[+] ions drive conformational selection. The rates of L-asp (**a**) and DL-TBOA (**b**) binding to P11W[IFS] as functions of ligand concentration. The rates of L-asp binding were measured in the presence of 0.5 (blue), 1 (red), 2 (green), 4 (purple) or 8 (orange) mM Na[+] ions. Data were fitted to *Equation 4* to determine plateau $k_r$ values (Materials and methods). DL-TBOA binding experiments were conducted in 4 (blue), 8 (red), 12 (green) or 16 (purple) mM Na[+] ions and fitted to *Equation 3* to obtain plateau $k_r$ values. All experiments were performed in triplicate and means and standard error bars are shown. (**c**) Schematic representation of Na[+]-driven conformational selection mechanism. In this model, T* is a conformation of the transporter that is unable to bind substrate with appreciable affinity. Na[+] binding is coupled to a transition to a conformation that binds L-asp and the remaining two Na[+] ions rapidly. (**d**) Na[+] dependence of $k_r$ values for L-asp (open squares) and DL-TBOA (solid squares) binding. Error bars are standard errors from fits in panels a) and b). The $k_r$ values for DL-TBOA at 20 and 24 mM were approximated by measuring $k_{obs}$ of 8 mM DL-TBOA binding (expected to be well at the plateau). Linear fits of L-asp (red) and DL-TBOA (blue) $k_r$-s were constrained to go through the origin, yielding slopes of 36 ± 0.9 M[-1] s[-1] and 26 ± 1.7 M[-1] s[-1].

DOI: https://doi.org/10.7554/eLife.37291.008

$$k_{obs} = \frac{k_{-r} + k_r + k_{off} + k_{on}[asp] - \sqrt{\left(k_{off} + k_{on}[asp] - k_{-r} - k_r\right)^2 + 4k_{-r}k_{on}[asp]}}{2} \qquad (4)$$

Most of the rate constants in the above equation are not resolved by the data, but $k_r$ values are well determined as they correspond to the plateaus reached at high ligand concentrations (see Materials and methods for details). The obtained $k_r$ values for L-asp binding also depended linearly on Na[+] concentration with a slope of 36 M[-1] s[-1], similar to that of DL-TBOA (*Figure 3d*). We note that while similar, the slope for DL-TBOA $k_r$ values is smaller than that for L-asp, and the linear fit does not pass through zero when unconstrained. This might be because at higher Na[+]

concentrations used in DL-TBOA binding experiments, $Na^+$ binding itself might become rate-limited by a $Na^+$-independent conformational transition.

## Kinetic model of coupled $Na^+$ and L-asp binding

Our data suggest that binding of the first $Na^+$ ion is rate limiting and drives a conformational transition of the transporter into a state competent to bind the remaining $Na^+$ ions and L-asp or DL-TBOA. Once the complex is fully formed, at least one $Na^+$ ion is able to rapidly equilibrate with the solution; its release is prerequisite for L-asp dissociation. The role of the third $Na^+$ ion remains ambiguous and it may contribute to both processes. Putting these considerations together, we have constructed the most parsimonious kinetic model compatible with the experimental data. The proposed model (*Figure 4a*) includes the initial slow $Na^+$ binding with the rate constant of 36 $M^{-1}$ $s^{-1}$, final $Na^+$ binding rapidly with a rate constant set arbitrarily to $10^7$ $M^{-1}$ $s^{-1}$, and two alternative branches reflecting the ambiguous role of the second $Na^+$ ion. The model contains twelve rate constants, which we estimated by fitting raw binding and dissociation data globally after fixing parameters for $Na^+$ binding (see Materials and methods for details). The constants that describe the rate limiting steps in the binding and dissociation reactions are well determined by the data. Other rate constants are only defined in terms of the lower limits (i.e., faster than the rate-limiting steps). The obtained rate constants (*Figure 4a* and *Figure 4—source data 1*) recapitulate well most of the kinetic data for L-asp binding and dissociation (*Figure 4—figure supplement 1a–c*). Only minor changes were required to reproduce DL-TBOA binding kinetics limited to about 20-fold increase in the rate of DL-TBOA unbinding (*Figure 4—figure supplement 2* and *Figure 4—source data 1*). As expected simulated data were in good agreement with experiment at lower $Na^+$ concentrations, but diverged somewhat at higher $Na^+$ concentrations. The fitted rate constants of L-asp binding and unbinding also yield L-asp equilibrium dissociation constant similar to experimental at 0.5 mM $Na^+$ (*Figure 4—figure supplement 1d*). However, the predicted $Na^+$ dependence of L-asp binding is steeper than the experimentally measured, and there is an increasing divergence between the predicted and measured constants at increased $Na^+$ concentrations. This discrepancy is likely due to our model predicting that the L-asp unbinding rates continue to decrease at increasing $Na^+$ concentrations (*Figure 3* and *Figure 4—figure supplement 1b*). It is possible, however, that an alternative disassembly mechanism becomes significant at higher $Na^+$ concentrations.

The relative fluxes through the two alternative reaction branches, 'second $Na^+$ ion binds first' (reactions 2 and 4 in *Figure 4a*) and 'L-asp binds first' (reactions 3 and 5), determine the shape of dependencies of $k_{obs}$ and $k_{obs,d}$ on $Na^+$ concentration and constrain the ratios of several rate constants (*Equations 7 and 8* in Materials and methods). Error analysis shows that excluding either of the branches from the fit leads to 20–40% increase in the sums of $\chi^2$, confirming that both branches contribute to the reaction. The flux through 'second $Na^+$ ion binds first' both during binding and dissociation of the complex is proportional to $Na^+$ concentration and is the predominant mechanism at high ion concentration (*Equation 7 and 8* and *Figure 4b*). However, the alternative route (reactions 3 and 5) becomes dominant at $Na^+$ concentrations below ~3.6 mM (*Figure 4b*). Therefore, both complex assembly and disassembly mechanisms should be described as probabilistic reaction paths rather than unique mechanisms. Regardless of the reaction path used, the three $Na^+$ ions and L-asp bind to P11W$^{IFS}$ in a concerted manner with no equilibrium or kinetic intermediates (*Figure 4c*). The disassembly of the complex also proceeds without intermediates at $Na^+$ concentrations of ~1 mM and above (*Figure 4c*). Only when $Na^+$ concentration approaches or falls below the dissociation constant for the rapidly equilibrating $Na^+$ ions, a kinetic intermediate TANa becomes populated.

## Increased rate of substrate binding to the 'fast mutant' of Glt$_{Ph}$

A potential concern with our experiments is the use conformationally constrained transporter mutants. To test whether cross-linking limits the rates of substrate gating, we examined binding to the so-called 'fast mutant' R276S/M395R (FM), which shows about four times faster turnover rate compared to the wild type Glt$_{Ph}$ (*Ryan et al., 2010*). The increased transport rate was attributed to the increased frequency of transitions of the transport domain from the outward to the inward facing position (*Akyuz et al., 2015*). The crystal structure of the mutant captured a distinct inward-facing conformation that was termed 'unlocked' because the transport domain leaned away from the scaffold, potentially providing space for HP2 to open (*Figure 5a*). To examine whether binding and

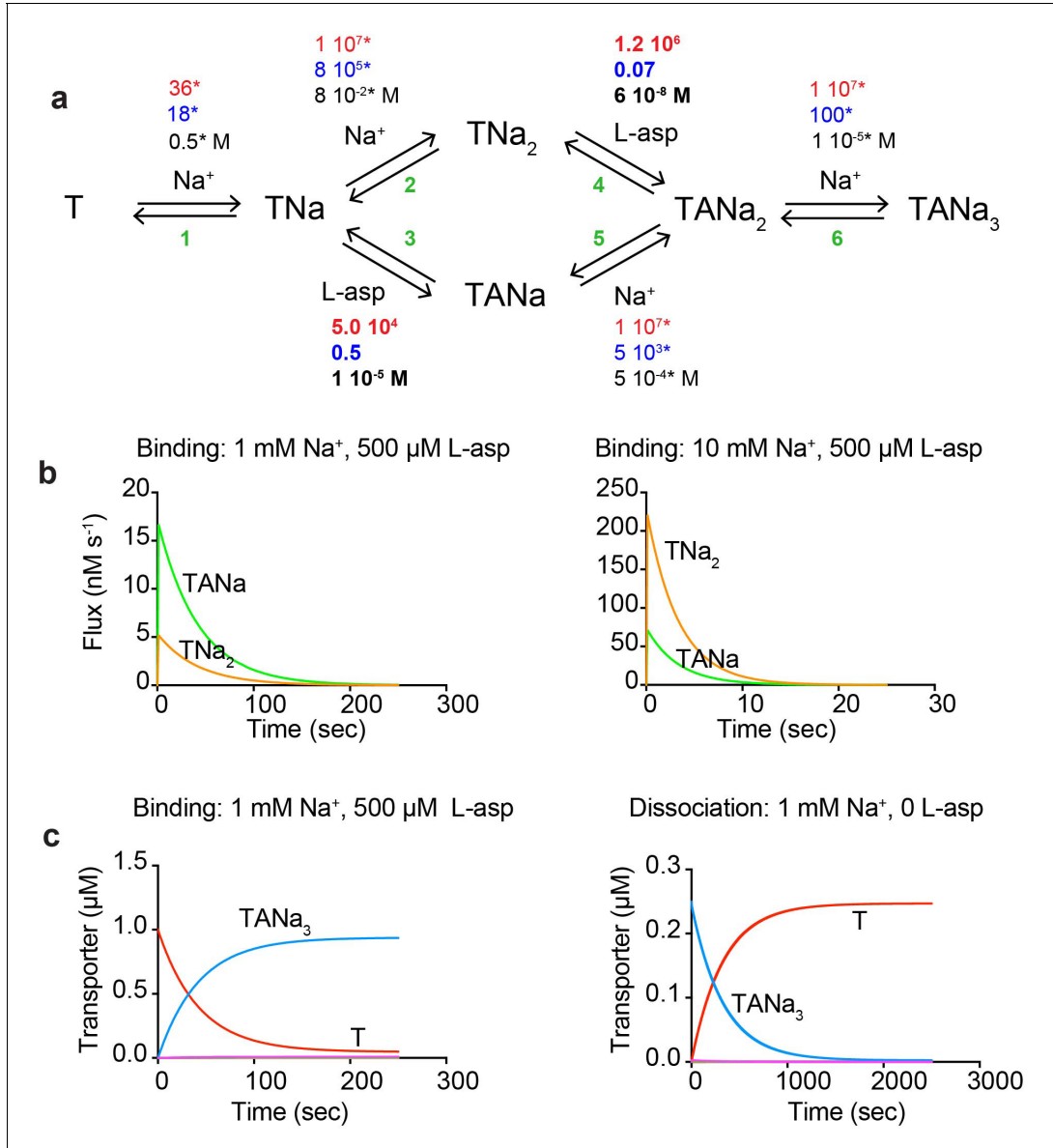

**Figure 4.** A kinetic model of $Na^+$ and L-asp binding to P11W$^{IFS}$. (**a**) The proposed reaction mechanism. The kinetic and equilibrium constants are shown next to the individual reactions (numbered in green). The on- (red) and off-rate constants (blue) are in $M^{-1} s^{-1}$ and $s^{-1}$, respectively. The equilibrium dissociation constants (black) are in M. Constants that were fixed during data fitting are marked by starts. (**b**) Simulated binding reaction, in which 1 μM apo transporter is mixed with $Na^+$ ions and L-asp. The amounts of ligands are indicated above the graphs that show reaction fluxes through 'second $Na^+$ ion binds first' (TNa$_2$, orange) and 'L-asp binds first' (TANa, green) reaction pathways. (**c**) Simulated binding and dissociation reactions showing that only apo transporter (T, red) and fully bound transporter (TANa$_3$, blue) become populated, while all potential reaction intermediates (magenta) are present only in miniscule amounts.

DOI: https://doi.org/10.7554/eLife.37291.009

The following source data and figure supplements are available for figure 4:

**Source data 1.** Kinetic rate constants summary.
DOI: https://doi.org/10.7554/eLife.37291.015
**Figure supplement 1.** Fitted rate constants describe experimental data well.
DOI: https://doi.org/10.7554/eLife.37291.010
**Figure supplement 2.** Experimental and simulated kinetics of L-asp binding to P11W$^{IFS}$.
DOI: https://doi.org/10.7554/eLife.37291.011
**Figure supplement 3.** Experimental and simulated kinetics of L-asp dissociation from P11W$^{IFS}$.
DOI: https://doi.org/10.7554/eLife.37291.012

*Figure 4 continued on next page*

Figure 4 continued

**Figure supplement 4.** Kinetic modeling of DL-TBOA binding to P11W[IFS].
DOI: https://doi.org/10.7554/eLife.37291.013
**Figure supplement 5.** Experimental and simulated kinetics of DL-TBOA binding to P11W[IFS].
DOI: https://doi.org/10.7554/eLife.37291.014

unbinding events are accelerated in this mutant, we introduced R276S/M395R mutations within P11W[IFS] background (FM-P11W[IFS]). We also determined the crystal structure of crosslinked K55C/ C321A/A364C/R276S/M395R $Glt_{Ph}$ (FM-$Glt_{Ph}$[IFS]). The structure was very similar to the structure of $Glt_{Ph}$[IFS], that is none of the protomers were in the 'unlocked' conformation and packed well against the scaffold (*Figure 5a* and *Table 1*). Nevertheless, the affinity of FM-P11W[IFS] for L-asp was similar to that of the unconstrained FM-$Glt_{Ph}$ (*Akyuz et al., 2015*) and about 100 fold weaker than that of the P11W[IFS]: 23 ± 7 and 0.23 ± 0.1 μM, respectively, at 10 mM Na[+]. Kinetics of L-asp binding to FM-P11W[IFS] revealed that $k_{obs}$ plateaued at elevated L-asp concentrations and that these plateaus were Na[+]-dependent, suggesting that the overall mechanism of L-asp and Na[+] binding is similar in FM-P11W[IFS] and P11W[IFS] (*Figure 5c*). The binding rates were about four times higher than those of P11W[IFS] at equivalent Na[+] ion concentrations. Our kinetic model described the data well. The key adjustments included the 4-fold increase of the rate of binding of the first Na[+] ion; modestly increased Na[+] affinity; and approximately thousand-fold increase of L-asp dissociation rates (accounting for lower affinity) (*Figure 5—figure supplement 1a* and *Figure 4—source data 1*). The reduced affinity and correspondingly increased dissociation rate might be due to distinct position of guanidinium group of R276 in the wild type transporter and R395 in FM (*Figure 5b* and *Figure 5— figure supplement 1a*). Computational studies showed that in the wild type transporter, R276 spends a significant proportion of the time hydrogen-bonded to D394, which coordinates the amino group of the bound L-asp. In FM protein, R395 does not show such propensity, perhaps explaining a more labile nature of L-asp complex (*Akyuz et al., 2015*).

The key conclusion from these experiments is that cross-linking does not limit the rate of the first Na[+] ion binding or the rate of substrate release. Furthermore, the increased rate of the first Na[+] ion binding in FM-P11W[IFS] suggests that the propensity to sample the unlocked conformation might increase the rate of the slow Na[+]-dependent gating event.

## Discussion

L-asp and Na[+] ions bind to $Glt_{Ph}$ cooperatively, with essentially no equilibrium intermediates in the lower Na[+] concentration range (*Reyes et al., 2013*), which mandates that Na[+] ions determine on- and/or off-rates of L-asp binding. Here we show that in the inward-facing state, Na[+] controls both on- and off-rates of the amino acid binding. Furthermore, we show that the slow low affinity binding of the first Na[+] ion is the rate-limiting step of the complex assembly, which is completed upon tight and rapid binding of the last Na[+] ion. Dissociation of one or two ions controls the kinetics of complex disassembly. Collectively, these features ensure that binding and unbinding of L-asp and Na[+] ions proceeds with no kinetic intermediates. Notably, when the transporter in the outward facing state is exposed to high extracellular Na[+] concentration in the absence of substrate Na[+]-only bound states will occur. Opening of HP2 tip, observed in the structure of Na[+]-only bound transporter may serve as an additional structural safeguard to prevent uncoupled Na[+] uptake (*Verdon et al., 2014*)

We do not have direct evidence to show that L-asp binding and dissociation in the outward-facing state follows the same kinetic mechanism. Indeed, the structural mechanism of gating in the inward-facing state has not yet been resolved. However, it seems likely that the energetic penalty paid by the first Na[+] ion to restructure the binding sites is similar, because the geometry of the sites is essentially identical in the two states both in the apo and fully bound states. Previous kinetic studies on unconstrained $Glt_{Ph}$ variants reported some of the same features that we have observed in the inwardly constrained transporter. In one study, a slow low affinity Na[+] binding was found to preceded binding of L-asp (*Hänelt et al., 2015a*). In the other, binding of two Na[+] ions was postulated to precede binding of the amino acid (*Ewers et al., 2013*). Interestingly, in this latter study the amino acid binding rates were also found to plateau at high concentrations. These plateaus occurred at distinct levels for L-asp, D-aspartate and L-cysteine sulfinic acid, suggesting that the slow step

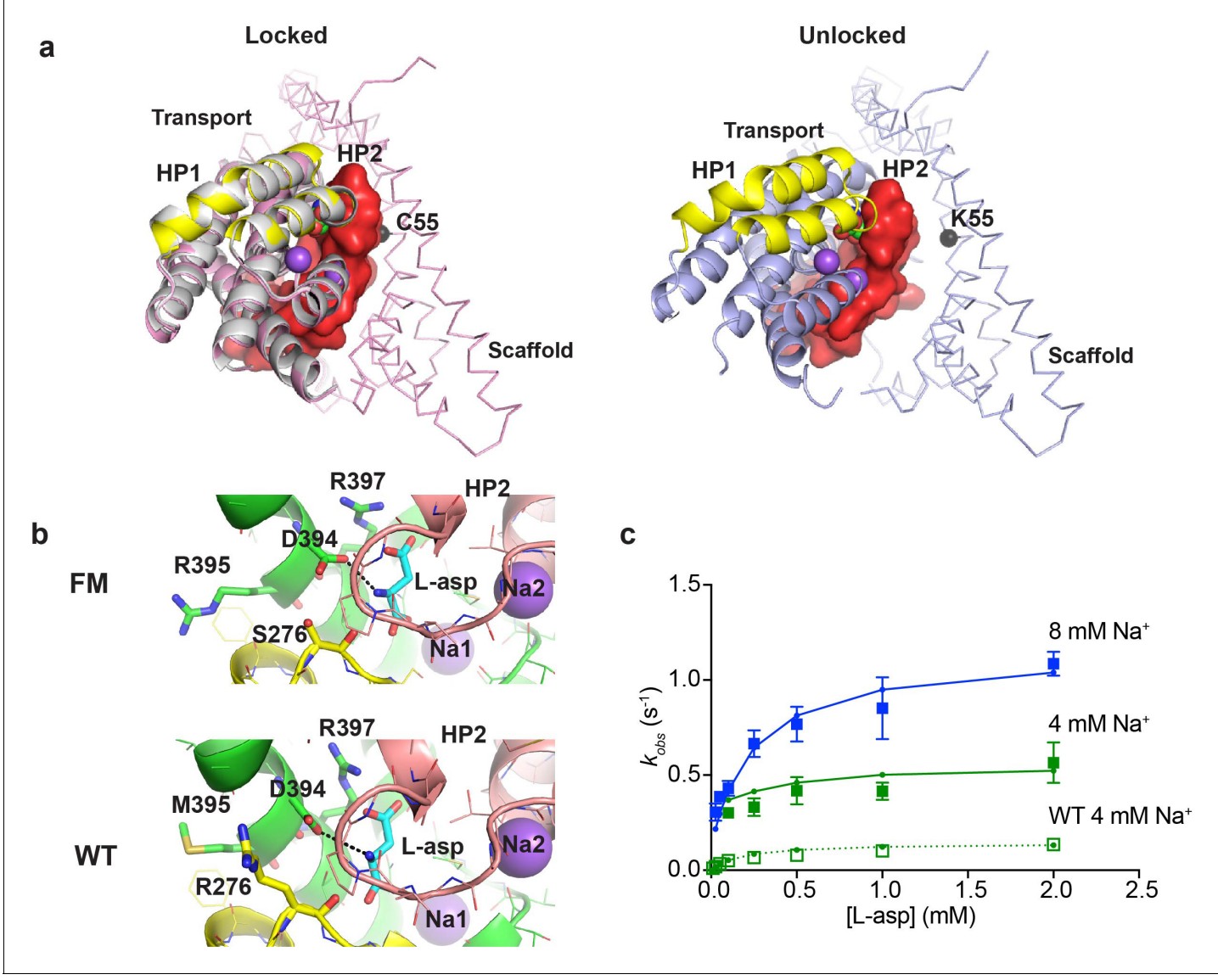

**Figure 5.** Crystal structure and L-asp binding kinetics of FM-Glt$_{Ph}$$^{IFS}$. (a) The superimposition of 'locked' conformations of FM-Glt$_{Ph}$$^{IFS}$ and Glt$_{Ph}$$^{IFS}$ (PDB code: 3KBC, left) and 'unlocked' conformation of FM-Glt$_{Ph}$ (PDB code: 4X2S, right). Single protomers are viewed from the cytoplasmic side of the membrane, rotated approximately 90°C compared to the representations in *Figure 1a* that are shown in membrane plane. FM-Glt$_{Ph}$$^{IFS}$, Glt$_{Ph}$$^{IFS}$ and FM-Glt$_{Ph}$ are colored pink, gray and blue, respectively. HP1 and HP2 of the FM variants are highlighted as yellow and red, respectively, and HP2 is shown in surface representation. A cleft is visible in uncross-linked FM-Glt$_{Ph}$, which may allow HP2 opening. Cα R.M.S.D. between FM-Glt$_{Ph}$$^{IFS}$ and Glt$_{Ph}$$^{IFS}$ structures is 0.4 Å. The Cα atom of residue 55, which is mutated to Cys and cross-linked to 364C in the transport domain of FM-Glt$_{Ph}$$^{IFS}$ is emphasized as a black sphere. (b) Close-up of the substrate-binding site in FM-Glt$_{Ph}$$^{IFS}$ (top) and Glt$_{Ph}$$^{IFS}$ (bottom). The key substrate-coordinating residues are highlighted as sticks, and the bond between the carboxyl group of D394 and amino group of the substrate is shown as a black dotted line. (c) Rate constants of L-asp binding to FM-P11W$^{IFS}$ in the presence of 4 (green) and 8 (blue) mM Na$^+$ ions. Solid lines through the data have been generated using the rate constant and kinetic model shown in *Figure 5—figure supplement 1*. Open green squares and dotted line show the rate constant of L-asp binding to P11W$^{IFS}$ in the presence of 4 mM Na$^+$ for comparison.

DOI: https://doi.org/10.7554/eLife.37291.016

The following figure supplements are available for figure 5:

**Figure supplement 1.** (a) The kinetic model describing L-asp binding to FM-P11W$^{IFS}$.
DOI: https://doi.org/10.7554/eLife.37291.017
**Figure supplement 2.** Experimental and simulated kinetics of L-asp binding to FM-P11W$^{IFS}$.
DOI: https://doi.org/10.7554/eLife.37291.018

**Table 1.** Crystallographic data and structure refinement

| | FM-Glt$_{Ph}$$^{IFS}$ |
|---|---|
| Data collection | |
| Space group | $C222_1$ |
| Cell dimensions | |
| a, b, c (Å) | 116.48, 196.42, 194.33 |
| a, b, g (°) | 90.00, 90.00, 90.00 |
| Resolution (Å) | 50.0–4.05 (4.19–4.05)* |
| $R_{sym}$ or $R_{merge}$ | 0.142 (0.982) |
| I / $\sigma_I$ | 17.8 (2.2) |
| Completeness (%) | 99.7 (99.7) |
| Redundancy | 12.2 (11.7) |
| Refinement | |
| Resolution (Å) | 19.9–4.05 |
| No. reflections | 18216 |
| $R_{work}$/$R_{free}$ | 0.2217/0.2660 |
| No. atoms | |
| Protein | 9138 |
| Ligand/ion | 27/6 |
| Water | 0 |
| B-factors | |
| Protein | 185.57 |
| Ligand/ion | 184.71/180.44 |
| R.m.s. deviations | |
| Bond lengths (Å) | 0.005 |
| Bond angles (°) | 0.989 |

*Values in the brackets correspond to the highest resolution shell.

DOI: https://doi.org/10.7554/eLife.37291.019

occurred after the initial low affinity binding, consistent with induced fit mechanism. In our studies, the maximal equilibration rates are Na$^+$-dependent and similar for L-asp and DL-TBOA, suggesting that the same Na$^+$ binding event occurs before binding of both ligands consistent with Na$^+$-dependent conformational selection mechanism. These findings are not contradictory, however. In our experiments, binding of the first Na$^+$ ion is rate limiting. Therefore, the detailed kinetic mechanism of L-asp binding to the transporter already bound to two Na$^+$ ions is not resolved, and may occur with additional steps that are all fast compared to the rate of binding of the first Na$^+$ ion.

Detailed kinetic schemes describing glutamate and Na$^+$ ions binding to EAATs have been proposed (*Watzke et al., 2001*; *Bergles et al., 2002*; *Mim et al., 2005*; *Zhou et al., 2014*). Most of them are consistent with the mechanism proposed here for Glt$_{Ph}$ in that one or two ions are believed to bind before glutamate and the remaining Na$^+$ ion(s) bind. Similarly Na$^+$ ion dissociation is thought to precede amino acid release. Such binding mechanism, in which both substrate binding and unbinding rates are controlled by Na$^+$ ions, provides significant physiological advantages. A transporter facing extracellular solution where Na$^+$ concentration is high will show rapid binding and slow release rate. One might envision that following glutamate-mediated neurotransmission, glutamate could bind to the transporters and remain bound for comparatively long times even as the concentration of the transmitter in and near the synapse falls below the dissociation constant. Thus, the slow release would be beneficial not only to increase the efficiency of transport, but also to retain glutamate bound to the transporter as its concentration drops to resting levels of ca. 25 nM, significantly below the transporter equilibrium affinity (*Chiu and Jahr, 2017*). Once the amino acid is translocated into the cell and the inward-facing transporter encounters low Na$^+$ concentration, glutamate

would be rapidly released. In contrast, an unbinding rate that is intrinsically slow and independent of $Na^+$ concentration would have resulted in slow release of the amino acid into the cytoplasm and low transport efficiency.

Structural interpretation of the kinetic data is difficult. Crystallographic, computational and mutational studies have located the three $Na^+$ binding sites in $Glt_{Ph}$ and closely related $Glt_{Tk}$ (*Boudker et al., 2007*; *Huang and Tajkhorshid, 2010*; *Bastug et al., 2012*; *Guskov et al., 2016*). Two sites, called Na1 and Na3, are located deep in the core of the protein below the substrate-binding site (*Figure 1—figure supplement 1*). The third $Na^+$ binding site, Na2, is at the surface of the protein between HP2 and the rest of the transport domain and is partially exposed to the solvent. In the structure of $Glt_{Ph}$ R397A mutant with drastically reduced substrate affinity that was crystallized in the presence of $Na^+$ ions only, the tip of HP2 is open and the Na2 site is unoccupied (*Figure 1—figure supplement 2*) (*Verdon et al., 2014*). Similarly, in $Glt_{Ph}$ variants bound to non-transportable blockers, including L-TBOA, HP2 is open and Na2 site unoccupied (*Boudker et al., 2007*; *Scopelliti et al., 2018*). Interestingly, in none of the three sites, the ions are coordinated by the bound L-asp directly; therefore all sites are coupled allosterically.

It is likely that the first $Na^+$ ion binds (step one in *Figure 4*) to a site with the highest intrinsic affinity, which according to computational studies is Na3 site, buried at the core of the transport domain (*Heinzelmann et al., 2011*; *Heinzelmann et al., 2013*). Its intrinsic affinity has been estimated at −16 kcal/mol. In contrast, experimental data suggest that the first $Na^+$ ion binds with affinity of no more than −0.5 kcal/mol. This large disparity is consistent with the observation that the geometry of all three $Na^+$ and substrate binding sites are drastically distorted in the apo protein (*Jensen et al., 2013*; *Verdon et al., 2014*). Thus, binding of the first $Na^+$ is likely slow and weak, because the protein undergoes significant remodeling, which must be energetically costly. Notably, this initial binding event is faster in the so-called fast mutant R276S/M395R variant FM-P11W[IFS], suggesting that the rate of the first ion binding is not limited by cross-linking used to conformationally constrain the transporter. Thus, the rate is likely controlled by the local restructuring events, even if it is affected by the propensity for the global transport domain movements. Once remodeling occurs, binding of the remaining ligands occurs much faster and with high affinity. It is difficult to establish unambiguously, which two $Na^+$-binding sites are vacated first during complex disassembly, but it seems likely that the surface Na2 site is one of them. In our experiments, the two 'locking' ions bind with high affinity around 100 µM. The tight binding appears to contradict computational studies that found $Na^+$ ion in Na2 site unstable or only weakly bound (*DeChancie et al., 2011*; *Heinzelmann et al., 2011*; *Zomot and Bahar, 2013*). The origin of this discrepancy is currently unclear.

In conclusion, we have examined the kinetic mechanism of binding and release of the substrate and coupled ions in the inward facing state of $Glt_{Ph}$. We show that the mechanism ensures lack of equilibrium and kinetic intermediates and that rates of both substrate binding and release are controlled by $Na^+$ ions.

## Materials and methods

### Mutagenesis, protein expression, and purification

All $Glt_{Ph}$ mutants were prepared by site-directed mutagenesis using cysteinless seven-histidine mutant of $Glt_{Ph}$ that has been used in several previous studies (*Reyes et al., 2009*; *Akyuz et al., 2013*; *Akyuz et al., 2015*). Proteins were expressed and purified as described previously (*Boudker et al., 2007*). In brief, $Glt_{Ph}$ mutants were expressed in *Escherichia coli* DH10b strain (Invitrogen, Inc). Small volumes of *E. coli* overnight cultures were diluted into Luria broth media supplemented ampicillin and grown at 37°C. Cultures were induced at optical density of 0.6 with 0.2% L-arabinose for 3 hr at 37°C. Cells were harvested by centrifugation and pellets frozen in liquid nitrogen and stored at −80°C until use. Thawed cells were disrupted by using cell disrupter (Avestin, Inc) at ~15,000 psi. Cell debris were removed by centrifugation at 7,700 *g* for 15 min at 4°C. Crude membranes were pelleted by ultracentrifugation at 180,000 *g* for 1 hr at 4°C. Membranes were homogenized in 20 mM Hepes/Tris pH 7.4, 200 mM NaCl and pelleted again by ultracentrifugation. Washed membranes were homogenized and then solubilized in the same buffer supplemented with 40 mM dodecyl-β,D-maltopyranoside (DDM) for 1 hr. Solubilized membrane were clarified by

ultracentrifugation at 164,000 $g$ for 1 hr at 4°C and incubated with pre-equilibrated Ni-NTA resin (Qiagen) with gentle shaking. Ni-NTA resin was washed with seven column volumes (CV) of 20 mM Hepes/Tris pH 7.4, 200 mM NaCl, 40 mM imidazole and 1 mM DDM. Proteins were eluted with 5 CV of 20 mM Hepes/Tris pH 7.4, 200 mM NaCl, 250 mM imidazole and 1 mM DDM. Eluted proteins were concentrated using concentrators with 100 kDa MWCO (Amicon). The tag was removed by thrombin digestion at room temperature overnight. Cleaved proteins were purified by size exclusion chromatography in 20 mM Hepes/Tris pH 7.4, 300 mM choline chloride and 1 mM DDM.

## Crosslinking

Before crosslinking, purified proteins were reduced with 5 mM TCEP at room temperature for 1 hr and purified by SEC. For crosslinking, 10-fold molar excess of $HgCl_2$ was added to protein solutions at concentrations below 1 mg/mL, and incubated for 15 min at room temperature. Crosslinked proteins to be used in binding assays were re-purified by SEC. Peak fractions were collected and concentrated. Protein concentration was measured using Nanodrop (Thermo-Fisher Scientific) using an extinction coefficient determined by ProtParam (*Gasteiger et al., 2005*). To check availability of free thiols after crosslinking, proteins were incubated with 5-fold molar excess of fluoroscein-5-maleimide (F5M). Fluorescent F5M-labeled proteins were imaged on SDS-PAGE under blue illumination and stained with Coomassie blue.

## Fluorescence-based kinetics assay

For emission scans, protein samples at 1 µM were prepared in 2 mL of 20 mM Hepes/Tris pH 7.4, 300 mM choline chloride and 1 mM DDM. Intrinsic protein fluorescence was excited at 295 nm and emission measured in bulk fluorometer (PTI Inc.). Baseline was determined using buffer only and subtracted from the data. For real-time fluorescence measurements of ligand dissociation, protein samples at 25 µM were prepared in 20 mM Hepes/Tris pH 7.4, 300 mM choline chloride, 10 mM NaCl, 0.5 mM L-asp 1 mM DDM and diluted 100 fold into the fluorometer cell containing the same buffer lacking L-asp and containing variable NaCl concentrations with fast stirring at 25°C. Data were normalized by the value of the fluorescence after addition of saturating $Na^+$/L-asp at the end of experiment. Approximately 10% reduction in protein fluorescence was observed upon dissociation. For real-time fluorescence measurements of ligand binding, protein samples at 1 µM were incubated in 2 mL fluorometer cell containing 20 mM Hepes/Tris pH 7.4, 300 mM choline chloride and 1 mM DDM with various NaCl concentrations at 25°C. Data were normalized by dividing fractional fluorescence change by the maximal change observed upon addition of saturating ligand concentrations at the end of each experiment. After signal equilibration, ligands were added with fast stirring. Excitation and emission wavelengths were 295 and 350 nm, respectively. Here and elsewhere, data were analyzed using SigmaPlot 12.0 (Systat Software, Inc) and GraphPad Prism 7 (GraphPad Software). All kinetic relaxation curves were fitted to single exponentials.

## Fluorescence-based titration using RH421

For L-asp titration, protein samples at 1 µM concentration were prepared in 20 mM Hepes/Tris pH 7.4, 300 mM ChoCl, and 0.4 mM DDM. After addition of appropriate NaCl concentrations and 200 nM RH421 (Sigma-Aldrich), the samples were incubated at 25°C until fluorescence signal plateaued. L-asp aliquots were added to the samples with fast stirring and fluorescence emission was recorded following equilibration. Excitation and emission wavelengths were 532 and 628 nm, respectively.

## Protein reconstitution into liposomes

$Glt_{Ph}$ protein samples were reconstituted into liposomes as previously described (*Boudker et al., 2007*; *Ryan et al., 2009*). In brief, 0.1% L-α-Phosphatidylethanolamine-N-lissamine rhodamine B sulfonyl (Avanti Polar Lipids) was added to with 3:1 (w/w) mixture of *E.coli* polar lipid extract and egg phosphatidylcholine and dried on rotary evaporator and under vacuum overnight. The mixture was hydrated using 20 mM Hepes/Tris, pH 7.4 and 200 mM KCl buffer at final lipid concentration of 5 mg/ml by 10 freeze/thaw cycles. The suspensions were extruded through 400 nm filter membranes (Whatman) 10 times to form unilamellar liposomes. The rhodamine fluorescence of the liposomes was measured to generate a concentration calibration curve; excitation and emission wavelengths were 560 and 583 nm, respectively. The liposomes were destabilized by addition of Triton X-100

0.5:1 (w:w) detergent to lipid ratio. Proteins were added to the destabilized liposome at a ratio of 1:1000 (w/w) of protein to lipid, and incubated for 30 min at room temperature. Detergent was removed by three rounds of incubation with 80 mg/ml of pre-washed SM-2 beads (Bio-Rad): 2 hr at room temperature, 2 hr at 4°C and overnight at 4°C with gentle agitation. The proteoliposomes were extruded 10 times through 400 nm membranes and concentrated by ultracentrifugation at 86,000 $g$ for 45 min at 4°C. To replace the internal buffer, proteoliposomes were re-suspended in 1 mL of appropriate buffer and subjected to three freeze/thaw cycles followed by extrusion. Final concentrations of proteoliposomes were determined by rhodamine fluorescence.

## Transport assays

For substrate exchange experiments, the internal proteoliposome buffer was replaced with buffer containing 20 mM Hepes/Tris, pH 7.4 and 200 mM NaCl and 100 µM L-asp. To initiate uptake or exchange, the proteoliposomes were diluted 1000 fold into reaction buffer containing 20 mM Hepes/Tris pH 7.4, 200 mM NaCl, and 500 nM $^3$H-L-asp (PerkimElmer, 13.2 mci/mmol) pre-incubated at 35°C. In uptake experiments external buffer was supplemented with 100 nM cold L-asp to match exchange buffer conditions. At appropriate time points, 200 µl aliquots were removed and diluted into 2 mL of ice-cold quench buffer containing 20 mM Hepes/Tris pH 7.4, 200 mM LiCl. Samples were immediately filtered through 0.22 µM filter membrane (Millipore) and washed with 3 mL of the quench buffer. The washed filter membranes were transferred into scintillation vials, and retained radioactivity was measured using LS6500 scintillation counter (Beckman Coulter). Background uptake was measured in the reaction buffer containing 20 mM Hepes/Tris pH 7.4, 200 mM KCl, 500 nM $^3$H-L-asp and 100 nM cold L-asp.

## Protein crystallization, diffraction data acquisition and processing

FM-Glt$_{Ph}$$^{IFS}$ was purified by size exclusion chromatography in 10 mM Hepes/Tris pH 7.4, 100 mM NaCl, and 7 mM n-decyl-ß-D-maltopyranoside (Anatrace). Peak fractions were pooled, concentrated and supplemented with 0.4 M NaCl, 0.1 M MgCl$_2$, and 1 mM L-asp. Crystallizations were performed using hanging drop vapor diffusion method by mixing protein solution with well solution containing 0.1 M ADA pH 7.0, 0.4 M NaCl, 30% PEG600 at 1:1 ratio. Protein crystals grew at 20°C. They were cryo-protected by soaking in the well solution supplemented with higher concentrations of PEG 600. Diffraction data were collected at Advanced Photon Source beamline 24-ID-C. Data were processed using HKL2000 and CCP4 suite (*Minor et al., 2000*; *Winn et al., 2011*). Molecular replacement was performed in PHASER (*McCoy et al., 2007*) using Glt$_{Ph}$ structure in the inward-facing state (PDB code: 3KBC) as a search model. The model from molecular replacement was further refined using REFMAC, PHENIX and COOT (*Emsley and Cowtan, 2004*; *Adams et al., 2010*) applying three-fold non-crystallographic symmetry. TLS was used during final refinement (*Winn et al., 2001*) (*Table 1*). We noted that while Hg$^{2+}$ was used to crosslink the protein, only traces of Hg$^{2+}$ ions could be observed in the structure, and disulfide bonds formed instead. The Hg$^{2+}$ ions appeared to be present at very low occupancy and we did not model them in the structure. All structural figures were prepared using Pymol (*Schrodinger, 2015*).

## Kinetic modeling and data fitting

To fit the dependencies of $k_{obs}$ at given Na$^+$ ion concentrations on concentrations of L-asp and DL-TBOA (*Figure 3a and b*) we need to consider the kinetic equation for the conformational selection mechanism (*Figure 3c*). The full *Equation 4* requires fitting of four rate constants, $k_r$, $k_{-r}$, $k_{on}$ and $k_{off}$ (*Vogt et al., 2012*). Of these, $k_r$ is the best-defined constant because $k_{obs}$ approaches $k_r$ asymptotically at high ligand concentrations. Notably, when $k_{off} > k_r$, as is likely the case for DL-TBOA binding, the values of $k_{obs}$ decrease with increased ligand concentration. Conversely, if $k_{off} < k_r$ then the values of $k_{obs}$ increase with increased ligand concentration as is observed for L-asp. The data for DL-TBOA binding were fitted to *Equation 3*. The data were fitted globally in Prism 7, keeping $k_{-r}$ value shared between datasets obtained at variable Na$^+$ concentrations, as it is not expected to be Na$^+$-dependent. Nevertheless, $k_{-r}$ and $K_{D,TBOA}$ were correlated and could not be resolved. The data for L-asp binding were fitted to the full *Equation 4*. The $k_{off}$ values were taken from the fit of the dissociation rates with Hill coefficient $n = 1$ (*Figure 2b*). The $k_{-r}$ and $k_{on}$ rate constants were correlated and could not be resolved, but $k_r$ values were well defined.

The kinetic models in *Figure 4*, *Figure 4—figure supplement 4* and *Figure 5—figure supplement 1* were implemented in KinTek program (*KinTek, Inc*) (*Johnson, 2009*) for global parameter fitting and in Copasi (*Hoops et al., 2006*) for simulations. For interactions between L-asp and P11W$^{IFS}$ protein, a 6-step kinetic model (*Figure 4*) with 12 rate constants was implemented. Raw relaxation curves, including 21 dissociation and 58 binding reactions were fitted globally in KinTek (*Figure 4—figure supplements 2* and *3*). The thermodynamic cycle through two alternative branches, 'second Na$^+$ binds first' (reactions 2 and 4 in *Figure 4*) and 'L-asp binds first' (reactions 3 and 5), was constrained. Additionally, the parameters were constrained to set the dissociation constant for the first two Na$^+$ ions (reactions 1 and 2 yielding $TNa_2$ complex) at 0.04 M$^2$ according to the previously measured apparent affinity of ~ 200 mM for Na$^+$ ions binding to Glt$_{Ph}$$^{IFS}$ in the absence of L-asp (*Reyes et al., 2013*). Specifically, the affinities for the first and second Na$^+$ ion were set at 0.5 M (reaction 1) and 0.08 M (reaction 2), respectively. These values reproduced experimentally measured Na + binding isotherms (*Reyes et al., 2013*). The combined affinity of the second and third Na$^+$ ions for TANa complex (reactions 5 and 6) was set to 5 10$^{-8}$ M$^2$ to fall within the range of apparent Na$^+$ affinities of 50–130 µM measured in L-asp dissociation experiments (*Figure 2b*). We noted that the quality of the fits was better if the second and third Na$^+$ bound cooperatively, that is the affinity for the third ion was higher than the second. Thus we set the affinities for the second (reaction 5) and third (reaction 6) Na$^+$ ions at 500 and 10 µM, respectively. The binding rate for the first Na$^+$ ion is well determined by the data and was fixed at 36 M$^{-1}$ s$^{-1}$ (*Figure 3c*). The remaining binding rates for Na$^+$ ions were set at an arbitrary high value of 10$^7$ M$^{-1}$ s$^{-1}$. Once binding parameters for Na$^+$ ions were set, the remaining rate constants of L-asp binding and unbinding were fitted with acceptable standard errors (*Figure 4—source data 1*). The total sum of $\chi^2$-s obtained during fitting was 4.7 10$^4$. Excluding 'L-asp binds first' branch of the reaction mechanism produced the best $\chi^2$ sum of 6.5 10$^4$ (an increase of 40%). Excluding 'second Na$^+$ binds first' branch produced best $\chi^2$ sum of 5.5 10$^4$ (an increase of 17%).

Notably, the flow of the reaction through the alternative branches determines the shapes of the $k_{obs}$ dependences on Na$^+$ concentrations in binding and dissociation reactions (*Figures 2* and *3*). In binding reactions, the probability to go through the 'second Na$^+$ binds first' branch (reactions 2 and 4) depends of Na$^+$ concentration as it is determined by the fraction of the transporter bound to the second Na$^+$, $F_{TNa2}$. Because the binding and unbinding rates of the second Na$^+$ ion to $TNa$ complex (reaction 2) are fast compared to the rates of L-asp binding to $TNa$ (reaction 3), we can consider this binding reaction in terms of rapid equilibrium. Thus, the fractions of $TNa_2$ and $TNa$ and will be

$$F_{TNa_2} = \frac{[Na^+]}{K_{D,2} + [Na^+]} \text{ and } F_{TNa} = \frac{K_{D,2}}{K_{D,2} + [Na^+]} \tag{5}$$

where $K_{D,2}$ is the dissociation contestant in reaction 2. The corresponding relative fluxes through $TNa_2$ and $TANa$ species will be:

$$\eta_{TNa_2} = F_{TNa_2} \cdot k_{on,4} \text{ and } \eta_{TANa} = F_{TNa} \cdot k_{on,3} \tag{6}$$

And the ratio between them

$$\frac{\eta_{TNa_2}}{\eta_{TANa}} = \frac{[Na^+]}{K_{D,2}} \frac{k_{on,4}}{k_{on,3}} \tag{7}$$

In dissociation reaction, the binding and unbinding of Na$^+$ ions to $TANa$ complex (reaction 5) can be considered in rapid equilibrium compared to L-asp unbinding in reaction 4. Therefore, similar reasoning can be applied as above, and fluxes through $TNa_2$ and $TNA$ species calculated as:

$$\frac{\rho_{TNa_2}}{\rho_{TANa}} = \frac{[Na^+]}{K_{D,5}} \frac{k_{off,4}}{k_{off,3}} \tag{8}$$

Importantly, these ratios (*Equations 7 and 8*), and not the individual equilibrium and rate constants, are defined by the experimental data. The fraction of the relative flux going through the 'second Na$^+$ binds first' branch relative to 'L-asp binds first' branch in both forward and reverse directions is proportional to Na$^+$ concentration with the equal fluxes through the two branches occurring at 3.6 mM Na$^+$. Simulations in Copasi confirmed these conclusions.

To fit DL-TBOA binding kinetic data, the same model was implemented, excluding binding of the third $Na^+$ ion (reaction 6). 68 individual relaxation curves were fitted globally, allowing only the rates of DL-TBOA dissociation to be adjusted (*Figure 4—figure supplement 5*). The best $\chi^2$ sum of 2 $10^4$ was obtained. Excluding 'DL-TBOA binds first' branch or 'second $Na^+$ binds first branch' produced similar $\chi^2$ values, indicating that these data alone are unable to distinguish between the models. Global fitting of 27 relaxation curves for FM-P11W[IFS] was accomplished in the same manner. The rates for L-asp dissociation and the rates for binding and unbinding of the first $Na^+$ ion were adjusted.

## Acknowledgement

This research used resources of the Advanced Photon Source, a U.S. Department of Energy (DOE) Office of Science User Facility operated for the DOE Office of Science by Argonne National Laboratory under Contract No. DE-AC02-06CH11357. We thank Narayanasami Sukumar for access to beamlines 24-ID-C at the Advanced Photon Source, and the staff for their assistance. The work has been supported by R01 NS064357 grant from National Institute of Neurological Disease and Stroke (OB). Structure of FM-Glt$_{Ph}$[IFS] has been submitted to PDB (accession code 6CTF). We thank Dr. Gerard Huysmans for critical reading of the manuscript.

## Additional information

### Competing interests

Olga Boudker: Reviewing editor, *eLife*. The other author declares that no competing interests exist.

### Funding

| Funder | Grant reference number | Author |
| --- | --- | --- |
| Howard Hughes Medical Institute | | Olga Boudker |
| National Institute of Neurological Disorders and Stroke | R01NS064357 | Olga Boudker |
| National Institute of Neurological Disorders and Stroke | R37NS085318 | Olga Boudker |

The funders had no role in study design, data collection and interpretation, or the decision to submit the work for publication.

### Author contributions

SeCheol Oh, Conceptualization, Data curation, Formal analysis, Validation, Investigation, Methodology, Writing—original draft, Writing—review and editing; Olga Boudker, Conceptualization, Resources, Data curation, Formal analysis, Supervision, Funding acquisition, Methodology, Writing—original draft, Project administration, Writing—review and editing

### Author ORCIDs

SeCheol Oh http://orcid.org/0000-0002-1685-5922
Olga Boudker https://orcid.org/0000-0001-6965-0851

### Decision letter and Author response

Decision letter https://doi.org/10.7554/eLife.37291.024
Author response https://doi.org/10.7554/eLife.37291.025

## Additional files

### Supplementary files

• Transparent reporting form

DOI: https://doi.org/10.7554/eLife.37291.020

### Data availability

Diffraction data have been deposited in PDB under the accession code 6CTF.

The following dataset was generated:

| Author(s) | Year | Dataset title | Dataset URL | Database, license, and accessibility information |
|---|---|---|---|---|
| Boudker O, Oh S | 2018 | Crystal structure of GltPh fast mutant - R276S/M395R | http://www.rcsb.org/structure/6CTF | Publicly available at the RCSB Protein Data Bank (accession no: 6CTF) |

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
