## [Decision Letter]

Thank you for submitting your article "Kinetic mechanism of coupling in sodium-aspartate symporter Glt_Ph_" for consideration by *eLife*. Your article has been reviewed by three peer-reviewers, and the evaluation has been overseen by José Faraldo-Gómez as Reviewing Editor and Richard Aldrich as the Senior Editor. The reviewers have opted to remain anonymous.

The reviewers have discussed the reviews with one another and the Reviewing Editor, who has drafted this decision to help you prepare a revised submission.

Summary:

This manuscript describes experiments designed to understand the molecular mechanism of Glt_Ph_, an archaeal sodium-aspartate symporter that is believed to be a model system for the Excitatory Amino Acid Transporters, or EAATs. The authors use tryptophan fluorescence to monitor the kinetics of L-aspartate (and DL-TBOA) binding to/release from a Glt_Ph_ construct locked in the inward-facing state, and to evaluate how these processes depend on [Na^+^]. This work builds on earlier studies from the Boudker lab, where ITC measurements were used to demonstrate that 3 Na^+^ ions and 1 aspartate bind to Glt_Ph_ with high cooperativity (Reyes, OH and Boudker, 2013). The authors had also proposed that this coupling is allosteric in nature and stems from Na^+^ binding events that precede substrate binding and that serve to 'remodel' the transporter core to a substrate-binding competent state (Reyes, OH, and Boudker, et al.2013). Here, the authors examine this mechanism further by quantifying the rates of association and dissociation of the protein/substrate/Na^+^ complex. They find that both the rates of substrate binding and dissociation are highly [Na^+^]-dependent. Specifically, the results presented confirm that binding of substrate requires prior occupancy by one or two Na^+^ ions. Based on experiments with DL-TBOA, the authors advance the idea that the initial binding of Na^+^, in the absence of substrate, permits the transporter to adopt a conformational state to which the substrate may bind with high affinity; that is, this process is one of conformational selection, rather than induced fit. Kinetic measurements also show that the aspartate off-rate depends on [Na^+^], implying that at least one Na^+^ ion binds rapidly after substrate recognition.

Having determined rate constants for all binding/dissociation steps for Na^+^ and substrate, the authors construct a kinetic model of the process, consistent with their measurements. The authors then simulate the time-evolution of the model, for different starting conditions. The authors observe that, under the conditions tested, the lifetime of all binding intermediates is negligible, i.e. only the apo- and fully loaded forms are populated, interconverting with each other. This observation leads to the authors to infer that this cooperativity in binding explains coupled transport in Glt_Ph_, as it would ensure that only the apo or fully occupied states of the transporter (3Na^+^:1Asp) have the chance to interconvert between outward- and inward-facing conformations.

Essential revisions:

The reviewers agree that the manuscript presents very interesting findings, and that the experimental work is designed thoughtfully and is of high quality.

However, the reviewers also agree that the authors' main claim – that the observed binding/unbinding cooperativity is sufficient to explain coupled transport (i.e. the absence of uncoupled transport) – is not convincingly supported by the data provided. The results reported in this manuscript and elsewhere (Reyes, OH and Boudker, et al.2013) demonstrate that one or two Na^+^ ions can bind to Glt_Ph_ in the absence of substrate, in both the outward- and inward-facing states. The same applies to EEATs; indeed, a Na^+^-loaded outer-facing conformation is probably highly populated in glutamatergic neurons between neurotransmitter-release events, under high extracellular [Na^+^] and little to no glutamate. Whether in physiological conditions or in functional assays, neither Glt_Ph_ or EAATs alternate between outward- and inward-facing conformations in this state bound to Na^+^ only – thereby precluding the uncoupled movement of Na^+^ and thus the dissipation of the Na^+^ electrochemical gradient. The kinetic cooperativity discussed in the manuscript is clearly significant, particularly in regard to the efficiency of the unloading of the complex in the inward-facing conformation, and thus the efficiency of the transport cycle. However, it does not explain why a Na^+^-bound transporter will not undergo the alternating-transition until it captures a molecule of substrate (and then one or more additional Na^+^ ions). This conformational restriction is of central importance for coupling the transport of substrate and Na^+^. The authors are therefore asked to amend the main narrative of the manuscript (title, Abstract, Discussion, etc.) to avoid unwarranted interpretations of the data, so that readers can better appreciate the mechanistic and physiological significance of these new observations.

A second important point raised by the reviewers concerns the mechanism by which the transporter adopts a state that is competent for substrate binding. The authors propose that this process is one of conformational selection, i.e. by binding the first Na^+^ ion the transporter would be able to access a high-affinity conformation, in the absence of substrate; when present, the substrate would then select that conformation, among others accessible but with low affinity. However, if this first Na^+^-binding step alone were sufficient to bring the transporter to the high-affinity state, then the value of K_D_(L-asp) in state TANa (K_D_ = 1x10^-5^ Figure 4A, reaction 3) would be the lowest achievable. However, this is not the case: e.g. K_D_(L-asp) in TANa2 state is K_D_ = 6x10^-8^, a couple orders of magnitude lower (Figure 4A, reaction 4). This suggests that further steps and conformational rearrangements – Na^+^-dependent or not (e.g. TNa2 + L-asp – TANa2 or TANa + Na^+^ – TANa2) – are required to achieve high affinity binding of aspartate. In view of these considerations, the authors should clarify what they mean by 'conformational selection' and state that there might be other steps/kinetic mechanisms that contribute to determining substrate affinity. In particular, the authors should elaborate on how their findings fit in with an earlier study (Ewers et al., 2013) that examined this same question and found that binding of substrate to Na^+^-loaded Glt_Ph_ proceeds through an induced fit mechanism (presumably, reaction TNa2 + L-asp – TANa2).

The following points should also be addressed to improve the clarity of the manuscript:

1) The authors should explain in the text why conformational selection causes the rate of approach to equilibrium to decrease. It's counterintuitive, but a very important point to distinguish conformational selection and induced fit mechanisms. (Perhaps by briefly describing how the expected time constants would change at the limits of zero and infinity substrate.)

2) It's not at all clear why the fast mutant is introduced at the end. Is the goal to test the kinetic scheme with a mutant with different kinetic behavior? Or to propose that the mechanistic basis for the 4-fold increase in transport is a 4-fold more rapid first-Na-binding event? As currently written, it seems like a digression (it's not really mentioned in the Abstract, Introduction or Discussion; unless a clearer rationale for investigating this mutant can be articulated, the manuscript might be better off without this section.

3) The reviewers require some indication that the actual equilibrium values in these assays are sensible (either by showing additional raw data or by a commenting on those values in the text). For example, for some of the data, the equilibrium values should be the same for all experiments, even as the rate of the approach to equilibrium changes (for example, data in Figure 2B, the fast dilution experiment, where all aspartate is expected to dissociate.) Is this what is observed?

---

## [Author Response]

Essential revisions:The reviewers agree that the manuscript presents very interesting findings, and that the experimental work is designed thoughtfully and is of high quality.

*However, the reviewers also agree that the authors' main claim – that the observed binding/unbinding cooperativity is sufficient to explain coupled transport (i.e. the absence of uncoupled transport) – is not convincingly supported by the data provided. The results reported in this manuscript and elsewhere (Reyes, OH and Boudker, 2013) demonstrate that one or two* Na^+^
*ions can bind to* Glt_Ph_
*in the absence of substrate, in both the outward- and inward-facing states. The same applies to EEATs; indeed, a* Na^+^*-loaded outer-facing conformation is probably highly populated in glutamatergic neurons between neurotransmitter-release events, under high extracellular [*Na^+^*] and little to no glutamate. Whether in physiological conditions or in functional assays, neither* Glt_Ph_
*or EAATs alternate between outward- and inward-facing conformations in this state bound to* Na^+^
*only – thereby precluding the uncoupled movement of* Na^+^
*and thus the dissipation of the* Na^+^
*electrochemical gradient. The kinetic cooperativity discussed in the manuscript is clearly significant, particularly in regard to the efficiency of the unloading of the complex in the inward-facing conformation, and thus the efficiency of the transport cycle. However, it does not explain why a* Na^+^*-bound transporter will not undergo the alternating-transition until it captures a molecule of substrate (and then one or more additional* Na^+^
*ions). This conformational restriction is of central importance for coupling the transport of substrate and* Na^+^*. The authors are therefore asked to amend the main narrative of the manuscript (title, Abstract, Discussion, etc.) to avoid unwarranted interpretations of the data, so that readers can better appreciate the mechanistic and physiological significance of these new observations.*

We agree with the reviewers that Na^+^ bound outward-facing transporters may exist under conditions when the transporter faces extracellular environment with high Na^+^ ion concentrations but lacking substrate. Therefore, mechanisms must exist to diminish Na^+^ leaks. Indeed, our previous structure has suggested that in the Na^+^-only bound outward facing Glt_Ph_, loop opening is observed that may prevent or significantly slow trans-membrane domain movement (*eLife*, 2014). It is notable, however, that the affinity for Na^+^ ions in the absence of the substrate is very low ranging in various measurements from 50-100 mM for the outward facing state and around 250 mM in the inward facing state. Thus, binding of the ions and the substrate while not completely cooperative is strongly coupled. To reflect the focus on the binding reaction we have changed the title and the abstract. We have also altered the Introduction and Discussion to avoid stating that no intermediates occur as they will likely occur under high Na^+^/no substrate conditions. We made this point explicit in the Discussion.

*A second important point raised by the reviewers concerns the mechanism by which the transporter adopts a state that is competent for substrate binding. The authors propose that this process is one of conformational selection, i.e. by binding the first* Na^+^
*ion the transporter would be able to access a high-affinity conformation, in the absence of substrate; when present, the substrate would then select that conformation, among others accessible but with low affinity. However, if this first* Na^+^*-binding step alone were sufficient to bring the transporter to the high-affinity state, then the value of* K_D_*(L-asp) in state TANa (*K_D_
*= 1x10^-5^Figure 4A, reaction 3) would be the lowest achievable. However, this is not the case: e.g.* K_D_*(L-asp) in TANa2 state is* K_D_
*= 6x10^-8^, a couple orders of magnitude lower (Figure 4A, reaction 4). This suggests that further steps and conformational rearrangements –* Na^+^*-dependent or not (e.g. TNa2 + L-asp – TANa2 or TANa +* Na^+^
*– TANa2) – are required to achieve high affinity binding of aspartate. In view of these considerations, the authors should clarify what they mean by 'conformational selection' and state that there might be other steps/kinetic mechanisms that contribute to determining substrate affinity. In particular, the authors should elaborate on how their findings fit in with an earlier study (Ewers et al., 2013) that examined this same question and found that binding of substrate to* Na^+^*-loaded* Glt_Ph_
*proceeds through an induced fit mechanism (presumably, reaction TNa2 + L-asp – TANa2).*

We thank the reviewers for pointing out the lack of clarity in our writing. Classically, conformational selection posits that the protein samples a minor conformation, which has high affinity for the ligand among other dominant conformation(s) with comparatively low affinities. As the ligand sequesters the competent conformation, progressively larger fraction of the protein isomerizes into the state. The rate at which the protein transitions into the binding-competent state limits the overall rate of binding (under the condition that binding/unbinding to the high affinity state is rapid). Overall, the larger is the quantity of the ligand added to the protein, the longer it takes to reach equilibrium as the larger portion of the protein needs to transition into the high affinity state. For this reason, the equilibration rate constants decrease at higher ligand concentrations.

In our case, the transporter that is not bound to Na^+^ is unable to bind L-asp with appreciable affinity, while the protein bound to the first Na^+^ ion is able to bind both L-asp and the second Na^+^ ion with comparatively high affinities. Notably, the fraction of the protein bound to the first Na^+^ ion is small because the affinity for this ion is low. Thus, as L-asp and the second Na^+^ ion sequester this small fraction of the transporter, increasing fraction of the protein binds the first Na^+^ ion. The overall behavior of the system resembles conformational selection, except that the rate of transitioning into the state capable to bind L-asp and the second Na^+^ ion is itself Na^+^-dependent, because the step involves binding of the first ion. To emphasize, it is not that Asp selects a high affinity state among the conformations accessible to the transporter bound to the first Na^+^ ion, but rather this ion-binding itself is coupled to the transition into the conformation capable of binding L-asp and additional Na^+^ ions with high affinity. Because binding of the first ion is slow, weak, and essential for the biding of other solutes, we hypothesize that it is associated with energetically costly conformational transition.

The relationship between L-asp and the second Na^+^ ion is less definitive: both are able to bind with appreciable affinities to the transporter pre-bound to the first ion even as they increase the affinity of each other. This additional increase of affinity might be due to additional smaller conformational changes or may reflect, for example, local favorable electrostatic interactions or minor rearrangements of side chains.

There is no contradiction between our findings and those of Ewer et al. In their experiments, Ewer et al. observe binding of L-asp to TNa2 state at nearly saturating Na^+^ concentrations. They observe that L-asp binding to this state is consistent with induced fit mechanism, i.e. L-asp binds first weakly to the state, and then progresses to tightly bound state in a rate-limiting step. In our experiments, we do not resolve such details of L-asp binding because the rates of complex assembly are limited by the rate of the first Na^+^ ion binding. The consequent steps are all comparatively fast and are not resolved in detail.

We have clarified the above points in the Discussion.

The following points should also be addressed to improve the clarity of the manuscript:1) The authors should explain in the text why conformational selection causes the rate of approach to equilibrium to decrease. It's counterintuitive, but a very important point to distinguish conformational selection and induced fit mechanisms. (Perhaps by briefly describing how the expected time constants would change at the limits of zero and infinity substrate.)

We have clarified the point (as discussed above) in the Results section on TBOA binding.

2) It's not at all clear why the fast mutant is introduced at the end. Is the goal to test the kinetic scheme with a mutant with different kinetic behavior? Or to propose that the mechanistic basis for the 4-fold increase in transport is a 4-fold more rapid first-Na-binding event? As currently written, it seems like a digression (it's not really mentioned in the Abstract, Introduction or Discussion; unless a clearer rationale for investigating this mutant can be articulated, the manuscript might be better off without this section.

We agree with the reviewers that the “fast mutant” is not well introduced and incorporated into the narrative. We investigated the kinetics of L-asp binding to the R276S/M395R mutant because we were concerned that cross-linking, which constraints the transporter in the inward facing state, might limit the rate of substrate binding and release. We observed that crosslinking of the fast mutant does not affect the equilibrium dissociation constant, which is approximately 100 fold higher than that of the wild type and that the overall binding rate of L-asp is about 4 fold faster. This observation implies that the unbinding rate must be ca 400 fold faster than that of the wild type. The main conclusion from this experiment is that cross-linking does not limit the rate of gating, and that those rates must be set by the local events in and near the substrate-binding site. At this point, we do not feel that mechanistic conclusions based on these data are warranted. Therefore, we do not feel that the results describing the fast mutant should be featured in the Abstract or Introduction. We have made changes to the Results section on the fast mutant to clarify the purpose of the experiments and expanded the Discussion to emphasize the point.

3) The reviewers require some indication that the actual equilibrium values in these assays are sensible (either by showing additional raw data or by a commenting on those values in the text). For example, for some of the data, the equilibrium values should be the same for all experiments, even as the rate of the approach to equilibrium changes (for example, data in Figure 2B, the fast dilution experiment, where all aspartate is expected to dissociate.) Is this what is observed?

To address the reviewers, concern, we have included examples of all raw data used in the analysis. Notably, the global fitting of binding and dissociation data for L-asp identifies kinetic and equilibrium constants that best describe all data collectively. How well they reproduce the data can be best judged from Figure 4—figure supplement 1. Overall, the values reproduce both binding (panels A and C) and dissociation (panel B) kinetics and reasonably well capture the experimental equilibrium dissociation constants determined in separate experiments (panel D). From the newly included Figure 4—figure supplements 2 and 3, one can draw the same conclusion.

For example, in raw dissociation experiments the same plateau is reached at all but the highest Na^+^ concentrations, where equilibration is expected to be very slow. We also included a typical experimental curve from a dissociation experiment. Here, following the completion of the dissociation reaction, we added high concentrations of Na^+^ and L-asp, a step that was necessary to estimate the initial protein fluorescence of an assembled complex.

As discussed above, in L-asp binding and dissociation kinetic experiments, there is little direct information on equilibrium binding because the majority of the reactions proceed to completion: in binding experiments, the transporter is fully bound at the end of the equilibration periods (except for the lowest Na^+^ and L-asp concentrations), and in dissociation, the substrate should be completely released. Thus, it is important and quite remarkable that the measured kinetic rates recapitulate binding equilibrium values that were determined separately. In TBOA binding experiments (Figure 4—figure supplements 4 and 5), it is possible to estimate binding affinities for TBOA from the plateaus reached at the end of the equilibration times. For example, at 8, 12 and 16 mM Na^+^, K_D_s for TBOA are ca 830, 400 and 180 µM, respectively. These values are reasonably consistent with those previously measure for L-TBA binding: ca 1000 and 300 µM at 10 and 16 mM Na^+^, respectively (Reyes, 2013).